# IL18 signaling promotes homing of mature Tregs into the thymus

**Cristina Peligero-Cruz[1], Tal Givony[1], Arnau Sebé-Pedrós[2,3], Jan Dobeš[1], Noam Kadouri[1], Shir Nevo[1], Francesco Roncato[1], Ronen Alon[1], Yael Goldfarb[1], Jakub Abramson[1]***

[1]Department of Immunology, Weizmann Institute of Science, Rehovot, Israel; [2]Centre for Genomic Regulation (CRG), Barcelona Institute of Science and Technology (BIST), Barcelona, Spain; [3]Universitat Pompeu Fabra (UPF), Barcelona, Spain

**Abstract** Foxp3+ regulatory T cells (Tregs) are potent suppressor cells, essential for the maintenance of immune homeostasis. Most Tregs develop in the thymus and are then released into the immune periphery. However, some Tregs populate the thymus and constitute a major subset of yet poorly understood cells. Here we describe a subset of thymus recirculating IL18R[+] Tregs with molecular characteristics highly reminiscent of tissue-resident effector Tregs. Moreover, we show that IL18R[+] Tregs are endowed with higher capacity to populate the thymus than their IL18R[−] or IL18R[−/−] counterparts, highlighting the key role of IL18R in this process. Finally, we demonstrate that IL18 signaling is critical for the induction of the key thymus-homing chemokine receptor – CCR6 on Tregs. Collectively, this study provides a detailed characterization of the mature Treg subsets in the mouse thymus and identifies a key role of IL18 signaling in controlling the CCR6-CCL20-dependent migration of Tregs into the thymus.

*For correspondence:
jakub.abramson@weizmann.ac.il

**Competing interests:** The authors declare that no competing interests exist.

## Introduction

Foxp3[+]CD25[+]CD4[+] regulatory T cells (Tregs) are a suppressive T cell subset, essential for the maintenance of immune tolerance and homeostasis (*Belkaid and Tarbell, 2009*; *Dominguez-Villar and Hafler, 2018*; *Josefowicz et al., 2012*; *Panduro et al., 2016*; *Sakaguchi et al., 2008*). In steady state, most Tregs remain in lymphoid organs. However, subsets of specialized Tregs have been described in various parenchymal tissues, including the muscle, lung, fat, skin, gut, or brain (*Ali et al., 2017*; *Arpaia et al., 2015*; *Burzyn et al., 2013*; *Feuerer et al., 2009*; *Ito et al., 2019*; *Tanoue et al., 2016*; *Whibley et al., 2019*). At these sites, Tregs perform functions beyond their conventional immunosuppressive roles, including regulation of tissue homeostasis, regeneration, and/or stem cell differentiation. For instance, Tregs were shown to promote tissue repair in multiple tissues, either by inhibiting detrimental inflammation, by promoting stem cell differentiation, or even by promoting epithelial proliferation (*Arpaia et al., 2015*; *Burzyn et al., 2013*; *D'Alessio et al., 2009*; *Dial et al., 2017*; *Dobaczewski et al., 2010*; *Dombrowski et al., 2017*; *Ito et al., 2019*; *Liesz et al., 2009*; *Mathur et al., 2019*; *Mock et al., 2014*; *Nosbaum et al., 2016*). Furthermore, in the skin, Tregs promote hair follicle stem cell proliferation and differentiation even in the absence of injury (*Ali et al., 2017*). Similarly in the gut, Tregs promote intestinal stem cell renewal (*Biton et al., 2018*), while in the bone marrow, Tregs maintain hematopoietic stem cell quiescence and pool size, support plasma B cells, and promote B cell differentiation (*Glatman Zaretsky et al., 2017*; *Hirata et al., 2018*; *Pierini et al., 2017*). In the adipose tissue, Tregs were even shown to control insulin sensitivity (*Bapat et al., 2015*; *Feuerer et al., 2009*) or metabolic response to cold (*Medrikova et al., 2015*).

The thymus represents the major source of Tregs in the body. For a long time, thymic Tregs were thought to represent a homogenous population, which, upon development, emigrates from the thymus and populates secondary lymphoid organs and/or parenchymal tissues. However, more recently, several independent studies reported that a considerable fraction of the thymic Treg pool consists of developmentally mature Tregs (hereafter referred to as 'mature Tregs') that have populated the thymus either through recirculation from the periphery or retention upon their maturation. Interestingly, these mature Tregs were shown to progressively increase with age, and to constitute the majority of the Treg pool in the aged thymus (*Bosco et al., 2006*; *McCaughtry et al., 2007*; *Thiault et al., 2015*; *Yang et al., 2014*; *Zhan et al., 2007*). Moreover, these Tregs show an activated and differentiated phenotype (*Thiault et al., 2015*) with intact suppressive capacity (*Yang et al., 2014*). Trafficking of those mature Tregs into the thymus occurs partly via CXCL12(SDF-1)/CXCR4 (*Thiault et al., 2015*) and CCL20/CCR6 (*Cowan et al., 2018*; *Cowan et al., 2016*) pathways. In particular, the CCL20/CCR6 axis has been implicated in homing of mature Tregs into the medullary area of the thymus, since medullary thymic epithelial cells (mTECs) were shown to be the major source of CCL20 (*Cowan et al., 2018*). The medullary compartment of the thymus has been previously shown to orchestrate negative selection of self-reactive thymocytes, as well as the de novo generation of Tregs (*Richards et al., 2015*). In fact, it has been described that Tregs that migrate back to the thymus (recirculating Tregs) inhibit de novo production of new Tregs via IL2 consumption (*Thiault et al., 2015*). Therefore, it is speculated that recirculating Tregs promote the age-dependent decline of Treg development. Despite all this progress, it is still unclear whether these mature Tregs represent a homogeneous population, what are their putative functional roles or what are the key signals that allow mature Tregs to populate the thymus under homeostasis and stress conditions.

In this article, we provide novel and important insights into the heterogeneity of mature Tregs in the thymus and demonstrate that they can be segregated into two distinct subsets (based on the expression of the receptor for IL18), which substantially differ in their molecular characteristics, kinetics, and resistance to stress- and age-induced thymus involution. Moreover, we further elucidate the molecular mechanisms that orchestrate the homing of mature Tregs into the thymus and highlight the importance of IL18R-signaling pathway in controlling the CCR6-CCL20-dependent migration of Tregs into the thymus.

## Results

### Expression of IL18R defines two major subsets of mature Tregs in the thymus

First, to better characterize 'mature' Tregs in the thymus we used *Foxp3*.RFP *Rag*.GFP mice, an in vivo reporter system allowing us to discriminate between 'newly produced' thymic Tregs and their developmentally more 'mature' counterparts. To this end, we crossed 'Treg reporter' mice expressing red fluorescent protein (RFP) under the control of forkhead box P3 (*Foxp3)* promoter (*Wan and Flavell, 2005*) with 'T cell development reporter' mice expressing green fluorescent protein (GFP) under the control of recombination-activating gene 2 (*Rag2*) promoter (*Yu et al., 1999*). Since *Rag2* transcription is terminated upon positive selection in the thymus and GFP half-life is ~56 hr; the expression of GFP serves as a useful marker of T cell maturity following their positive selection (*McCaughtry et al., 2007*). Indeed, according to previous studies, T cells that have lost GFP expression are at least 3 weeks old (*Boursalian et al., 2004*). Therefore, these double reporter mice are a useful tool for the identification and discrimination of the 'newly produced' Tregs (*Foxp3*.RFP$^+$ *Rag*.GFP$^+$) and the more 'mature' Tregs (*Foxp3*.RFP$^+$ *Rag*.GFP$^-$), which have either been retained within the thymus or have recirculated there from the periphery (*McCaughtry et al., 2007*; *Thiault et al., 2015*; *Yang et al., 2014*). Alternatively, we took advantage of differential expression of CD73, which was previously shown to efficiently discriminate between 'newly produced' Tregs (CD73$^-$) and 'mature' Tregs (CD73$^+$) in the thymus (*Owen et al., 2019*), and showed strong negative correlation to *Rag*.GFP reporter expression (*Figure 1—figure supplement 1*).

To molecularly characterize mature Tregs in the thymus, we performed RNA sequencing on *Rag*.GFP$^-$ and *Rag*.GFP$^+$ Tregs that were isolated from the thymus and Tregs isolated from the spleen. In agreement with a previous report (*Thiault et al., 2015*), we found that mature Tregs from the

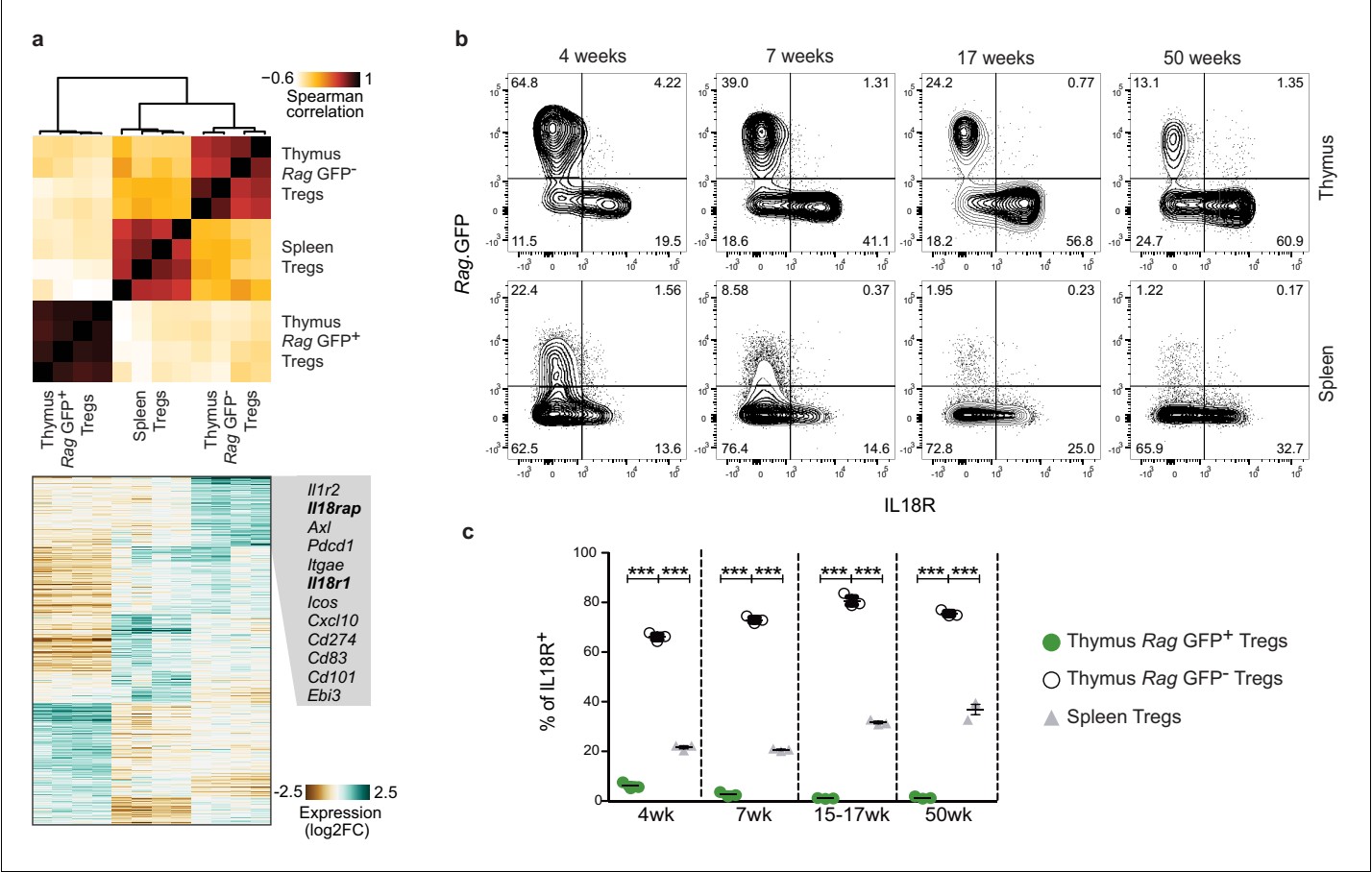

**Figure 1.** A subpopulation of mature Tregs in the thymus expresses IL18R. (a) Top, a heatmap showing Spearman's correlation coefficient between different samples based on the expression of differentially expressed genes. Samples include four biological replicates each of *Rag*.GFP⁻ Tregs from the thymus, Tregs from the spleen, and *Rag*.GFP⁺ Tregs from the thymus. Bottom, a heatmap showing the normalized expression of variable genes across samples. Specific genes upregulated in *Rag*.GFP⁻ Tregs are highlighted. (b) Representative flow cytometry plot showing the expression of IL18R in Tregs from the thymus (up) and the spleen (down). Numbers indicate the percentage of cells within each gate. (c) Frequency of IL18R⁺ cells among 'newly produced' *Rag*.GFP⁺ Tregs in the thymus (green dots), 'mature' *Rag*.GFP⁻ Tregs in the thymus (white dots) and Tregs in the spleen (gray triangles) in mice from different ages. Each dot represents an individual mouse. The mean ± SEM (standard error of the mean) is shown. Data are from one representative experiment with three biological replicates. Significant differences were determined using a 2-way ANOVA, corrected for multiple comparisons by the Bonferroni method and indicated by asterisks ***$p<0.001$.

The online version of this article includes the following figure supplement(s) for figure 1:

**Figure supplement 1.** Treg gating strategy.

**Figure supplement 2.** Mature (*Rag*.GFP⁻ or CD73⁺) Tregs are intrathymic (not a contamination from the bloodstream).

thymus are transcriptionally closer to Tregs from the spleen than to their *Rag*.GFP⁺ thymic counterparts (*Figure 1a*). Specifically, mature Tregs from the thymus differentially expressed genes related to immune activation (*Axl, Pdcd1, Cxcl10, Cd274, Cd101,* and *Ebi3*), genes characteristic of the recently described non-lymphoid tissue-like Tregs in lymph nodes (*Il1r2, Itgae, Icos,* and *Cd83*) and genes characteristic of non-lymphoid tissue Tregs including *Il18r1* and *Il18rap*, which encode for the IL18R complex (*Arpaia et al., 2015*; *Levine et al., 2014*; *Miragaia et al., 2019*; *Panduro et al., 2016*). The marked differences between the expression of *Il18r1* and *Il18rap* in *Rag*.GFP⁺ vs. *Rag*.GFP⁻ Treg was of interest, as IL18R⁺ Tregs in the lung have been shown to play a major role in tissue repair (*Arpaia et al., 2015*). To validate the expression of IL18R protein at a single cell resolution we performed flow cytometry analysis. This analysis confirmed that a large fraction (60–80%) of mature (*Rag*.GFP⁻) Tregs from the thymus expressed IL18R, while only a minor fraction of newly produced (*Rag*.GFP⁺) Tregs (1–8%) or Tregs from the spleen (20–40%) expressed IL18R in young, adult, or aged mice (*Figure 1b–c*). By both intravascular staining and perfusion techniques, we validated that

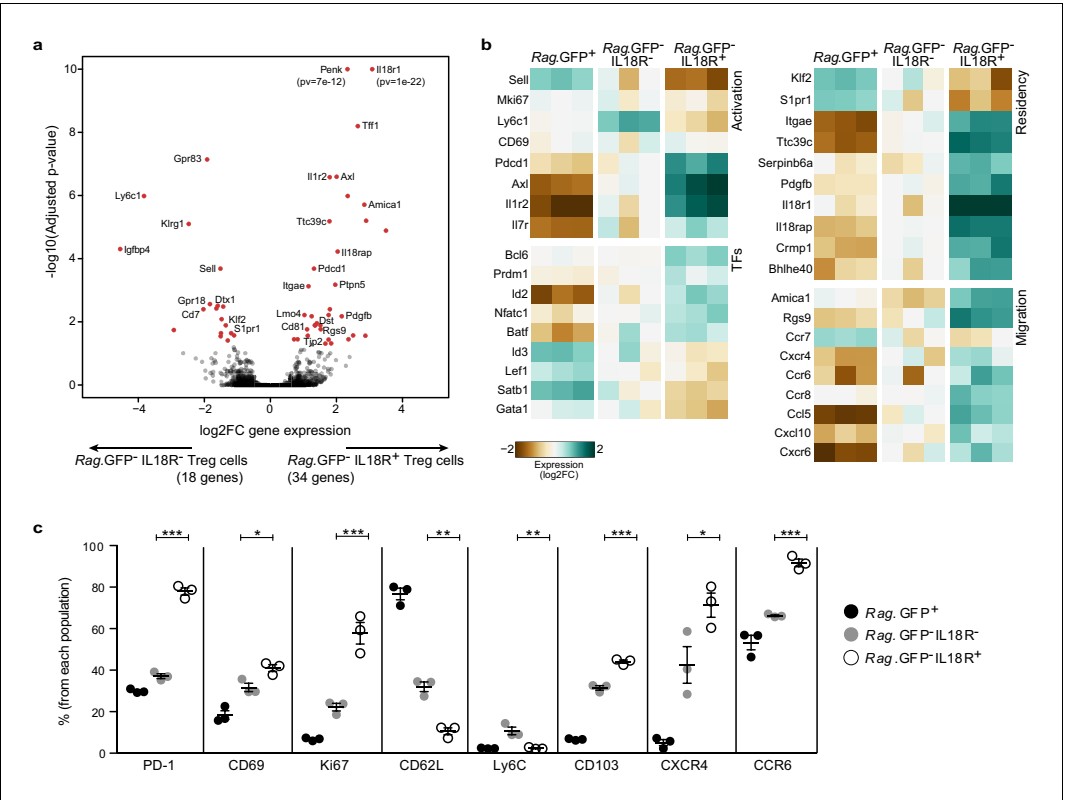

**Figure 2.** IL18R⁻ and IL18R⁺ represent molecularly distinct subsets of mature Tregs in the thymus. (**a**) A volcano plot showing differential gene expression between thymic *Rag*.GFP⁻IL18R⁻ Tregs and *Rag*.GFP⁻ IL18R⁺ Tregs. Genes in red have an adjusted p-value <0.05. Y axis is cut at 1e-10 for visualization. (**b**) A heatmap showing the normalized expression of selected variable genes in *Rag*.GFP⁺ Tregs, *Rag*.GFP⁻IL18R⁻ Tregs, and *Rag*.GFP⁻IL18R⁺ Tregs from the thymus. (**c**) Frequencies of different markers analyzed using flow cytometry in *Rag*.GFP⁺ Tregs, *Rag*.GFP⁻IL18R⁻ Tregs, and *Rag*.GFP⁻IL18R⁺ Tregs from the thymus. Each dot represents an individual mouse. The mean ± SEM (standard error of the mean) is shown. Data are from one experiment with three biological replicates. Significant differences were determined using ANOVA, corrected for multiple comparisons by the Bonferroni method and indicated by asterisks *p<0.05, **p<0.01, ***p<0.001.

The online version of this article includes the following figure supplement(s) for figure 2:

**Figure supplement 1.** RNA Sequencing protein validation, Representative gating from *Figure 2c*.

**Figure supplement 2.** E-cadherin (the ligand for CD103) is expressed by thymic epithelial cells.

---

mature Tregs do not represent contamination from circulating Tregs in the bloodstream (*Figure 1— figure supplement 2*). Therefore, this analysis uncovered two populations of mature Tregs (*Rag*. GFP⁻) in the extravascular space of the thymus, which differ by their expression of IL18R (gating in *Figure 1—figure supplement 1a*).

## IL18R⁻ and IL18R⁺ Tregs represent molecularly distinct subsets of mature Tregs in the thymus

To delineate potential differences between IL18R⁺ and IL18R⁻ mature Treg populations in the thymus, we isolated these populations based on their *Rag*.GFP reporter expression and subjected them to bulk RNA sequencing analysis (*Figure 2a–b*). In parallel, we used flow-cytometry to further validate our data (*Figure 2c*, *Figure 2—figure supplement 1*). Indeed, results of these analyses validated that these populations represent two molecularly distinct subsets, which differ in the expression of several key genes related to T cell activation, residency, and migration.

First, the IL18R⁺ Tregs, as opposed to IL18R⁻ Tregs, showed a more activated phenotype. Notably, the IL18R⁺ Treg subset expressed higher levels of T cell activation-associated markers including *Il1r2*, *Axl*, *Pdcd1* (PD-1), CD69, and Ki67; and lower *Sell* (CD62L) and *Ly6c1* (Ly6c), both known to be

downregulated in activated Tregs (*Delpoux et al., 2014*; *Lee et al., 2018*). In addition, IL18R$^+$ Tregs showed a transcription factor signature characteristic of effector Tregs, including elevated expression of *Bcl6, Prdm1, Nfatc1, and Batf* (*Cretney et al., 2011*; *Hayatsu et al., 2017*; *Levine et al., 2014*; *Sawant et al., 2012*; *Vasanthakumar et al., 2015*) and reduced expression of *Lef1, Satb1,* and *Gata1* (*Yang et al., 2019*). Similarly, the higher expression of *Id2* and lower *Id3* is also consistent with IL18R$^+$ Tregs being more terminally differentiated (*Yang et al., 2011*).

Second, the IL18R$^+$ Tregs, as opposed to IL18R$^-$ Tregs, were characterized by differential expression of several genes that are linked with tissue-resident populations. Specifically, the IL18R$^+$ Treg subset expressed lower levels of *Klf2* and its target *S1pr1*, which are essential for the egress of thymocytes from the thymus into the immune periphery (*Carlson et al., 2006*; *Matloubian et al., 2004*) and appear to be universally downregulated in tissue-resident populations (*Mackay and Kallies, 2017*). In addition, IL18R$^+$ Tregs expressed higher levels of *Itgae* encoding the αE integrin (CD103), which is characteristic of tissue-resident memory T cells and intraepithelial T cells (*Cepek et al., 1994*; *Suffia et al., 2005*). In the thymus, CD103 may possibly mediate retention of IL18R$^+$ Tregs through binding to its natural ligand E-cadherin, which is highly expressed by thymic epithelial cells (*Figure 2—figure supplement 2*). In addition, IL18R$^+$ Tregs expressed several genes that are characteristic of muscle/tissue-resident Tregs, including elevated expression of *Ttc39c, Serpinb6a, Pdgfb, Pdcd1, Il18r1, Il18rap, Penk, Crmp1, Bhlhe40, Axl* and reduced expression of *Ly6c1, Pdlim1, Gpr18, Dtx, Sell, S1pr1, Igfbp4, Gpr83, Cd7* (*Burzyn et al., 2013*), further supporting that the IL18R$^+$ Tregs represent a bona fide thymus-resident population.

Third, IL18R$^+$ Tregs expressed elevated levels of several genes linked with cell migration, suggesting that they might have recirculated from the periphery. Specifically, IL18R$^+$ Tregs were characterized by higher expression of A*mica1* and *Rgs9*, which are associated with activation and migration (*Agenès et al., 2005*; *Bazzoni, 2003*; *Ebnet et al., 2004*). In addition, IL18R$^+$ Tregs, compared with IL18R$^-$ Tregs, expressed lower *Ccr7* and higher *Ccr8, Cxcr6, Cxcr4,* and *Ccr6*, as well as higher protein levels of CXCR4 and CCR6, two main chemokine receptors that have been previously proposed to mediate recirculation of Tregs into the thymus (*Cowan et al., 2016*; *Thiault et al., 2015*).

In summary, the data indicate that IL18R$^+$ Tregs are a distinct population of effector Tregs that might have recirculated from the periphery and might reside in the thymus; while IL18R$^-$ Tregs (*Rag.*GFP$^-$IL18R$^-$) constitute an intermediate population between the newly produced Tregs (*Rag.*GFP$^+$) and the mature highly activated IL18R$^+$ Tregs (*Rag.*GFP$^-$IL18R$^+$).

## IL18R$^+$ Tregs are resistant to age- and stress-dependent thymus involution

To gain insight into the biology of both IL18R$^-$ and IL18R$^+$ mature Tregs in the thymus, we next investigated whether the abundance and/or frequency of these populations may change upon age-dependent or stress-induced thymic involution. For this purpose, we first analyzed the frequencies and numbers of the corresponding Treg subsets in thymi isolated from 4- to 52-week-old mice. As expected, the entire CD4$^+$ SP compartment, including the newly produced (*Rag.*GFP$^+$) Tregs progressively declined with age, mirroring the natural age-dependent thymus involution (*Figure 3a–c*). Interestingly, however, while the mature (*Rag.*GFP$^-$) IL18R$^-$ Tregs also progressively declined with age, the mature IL18R$^+$ Tregs persisted. Interestingly, the IL18R$^+$ Tregs transiently peaked around week 7 of age, suggesting that this could be linked to increased sex hormone levels when reaching sexual maturity. This seems in line with previous studies showing that sex hormones may potentiate *Il18* expression (*Kusumoto et al., 2005*). Then, IL18R$^+$ Tregs remained quite constant from week 10 to week 50, despite the dramatic decline of the thymic mass (*Figure 3d* and *Figure 3—figure supplement 1*). These data, therefore, demonstrate that the mature IL18R$^+$ Tregs constitute the most predominant Treg fraction in the aged thymus as their numbers remain stable (i.e. their frequency increases) over time (*Figure 3—figure supplement 1*).

Next, we analyzed the thymic Treg compartment upon stress-induced thymic involution. For this purpose, we utilized CD73 to identify mature Tregs (CD73$^+$) from newly produced Tregs (CD73$^-$) in the thymus as previously described in *Owen et al., 2019* (*Figure 1—figure supplement 1*). This setting allowed us to use enough aged-matched mice for stress and control conditions. We evaluated the numbers and frequencies of different Treg subsets in the thymus under different stressors (*Figure 4*). Those included 425cGy irradiation and 20 mg/kg of dexamethasone (two treatments commonly used to provoke acute thymic involution). Interestingly, both irradiation (*Figure 4a*) and

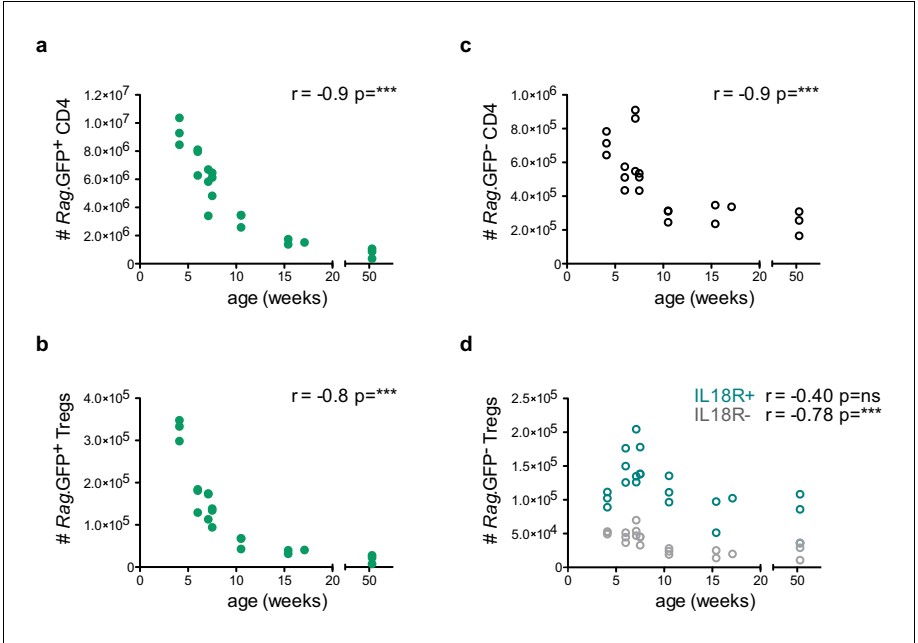

**Figure 3.** IL18R[+] Tregs are resistant to age-dependent thymus involution. Thymocyte counts as a function of age in weeks. (**a**) *Rag*.GFP[+] CD4 conventional T cells, (**b**) *Rag*.GFP[+] Tregs, (**c**) *Rag*.GFP[-] CD4 conventional T cells, and (**d**) *Rag*.GFP[-] IL18R[-] Tregs (in gray) and *Rag*.GFP[-] IL18R[+] Tregs (in blue). Each symbol represents an individual mouse. Data are from two independent experiments with 21 total mice. The Spearman's rank correlation coefficient (r) and p-value (p) for the correlations between cell counts and mice age are indicated by asterisks ***p<0.001, ns: non-significant.

The online version of this article includes the following figure supplement(s) for figure 3:

**Figure supplement 1.** Frequency and counts of IL18R[+] *Rag*.

dexamethasone (**Figure 4b**) treatments significantly increased the frequency of mature IL18R[+] Tregs, but not of the other Treg subsets. Correspondingly, both types of treatment also resulted in a substantial depletion of both the newly produced CD73[-] and the mature CD73[+] IL18R[-] Treg populations, whose numbers declined by >95% and >80%, respectively. In large contrast, the numbers of CD73[+] IL18R[+] Tregs decreased only by twofold in response to irradiation (**Figure 4a**) and remained unchanged in response to dexamethasone-induced involution (**Figure 4b**). The observed stability of IL18R[+] Tregs under age- and stress-dependent involution suggests that IL18R[+] Tregs represent a thymus-resident population that is resistant to stress-induced changes in the tissue and may thus be involved in the regulation of thymus homeostasis.

## IL18R contributes to maintain mature Tregs in the thymus

Based on the data reported above, we next wondered whether the IL18R[+] Tregs can migrate into the thymus and whether IL18R contributes to their retention in the thymus. For this purpose, we isolated B220-, CD8-, and IL18R- magnetically depleted splenocytes from CD45.2 *Foxp3*.GFP donors or B220- and CD8- magnetically depleted splenocytes from CD45.2 *Foxp3*.RFP donors and co-transferred them into CD45.1 recipients (**Figure 5a**, **Figure 5—figure supplement 1**). This allowed us to conveniently monitor the fate of GFP[+] Tregs containing only IL18R[-] population vs. RFP[+] Tregs containing both IL18R[-] and IL18R[+] populations within the same recipient. Indeed, 12 days post-injection, both GFP[+] and RFP[+] injected Tregs were detected in the spleens of the recipients demonstrating that the cells were transferred successfully (**Figure 5—figure supplement 2**). Interestingly, some of the injected GFP[+] Tregs (which originally contained only IL18R[-] Tregs) expressed IL18R, suggesting that some IL18R[-] Tregs gave rise to IL18R[+] Tregs already in the periphery (**Figure 5—figure supplement 3**).

Interestingly, while the transferred RFP+ Tregs successfully populated the recipients' thymi, only a small fraction (~20-fold less) of GFP[+] Tregs was detected in the thymus of the recipient mice.

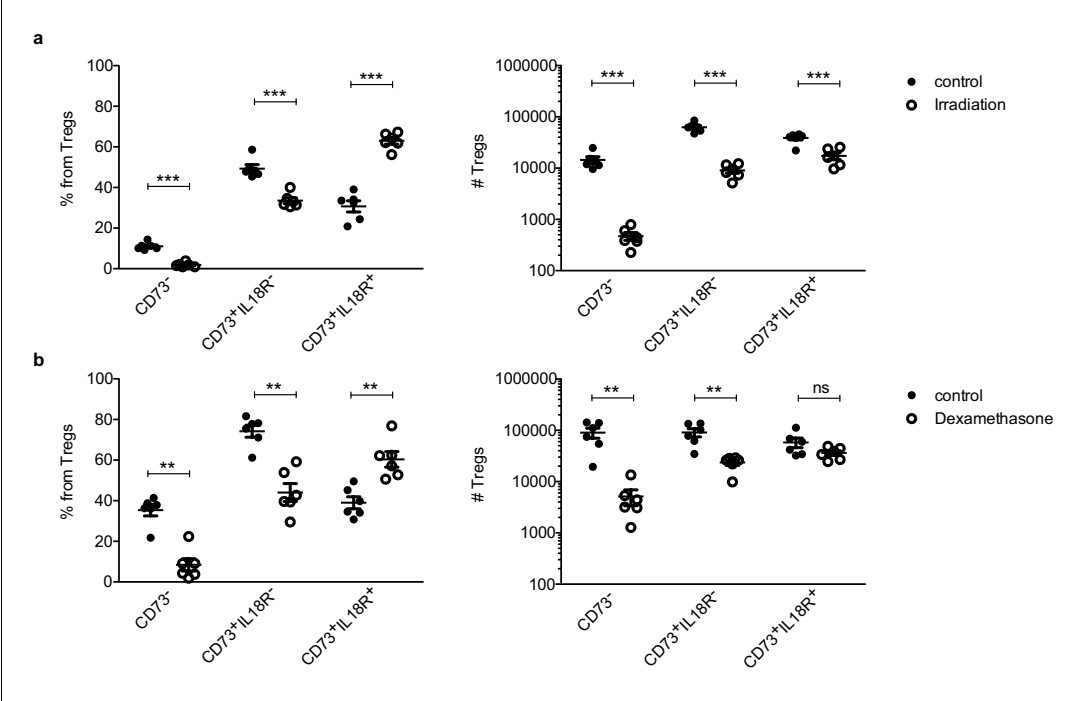

**Figure 4.** IL18R⁺ Tregs are resistant to stress-dependent thymus involution. Frequencies (left) and counts (right) of 'newly produced' CD73⁻ Tregs, 'mature' CD73⁺IL18R⁻ Tregs and 'mature' CD73⁺IL18R⁺ Tregs in control mice (solid dots) and treated mice (empty dots) upon different perturbations: (a) 7 days after 425cGy irradiation and (a) 7–10 days after 20 mg/kg dexamethasone injection. Each dot represents an individual mouse. The mean ± SEM (standard error of the mean) is shown. Data in (a,b) are from a representative experiment with six biological replicates. Significant differences were determined using 2-way ANOVA, corrected for multiple comparisons by the Bonferroni method and indicated by asterisks **p<0.01, ***p<0.001, ns: non-significant.

Indeed, the ratio of RFP⁺ to GFP⁺ Tregs was substantially higher in the thymus compared with that of the spleen, suggesting that IL18R⁺ Tregs have a higher capacity to recirculate into the thymus than their IL18R⁻ Treg counterparts (*Figure 5b*, *Figure 5—figure supplement 2*). In addition, the frequency of IL18R⁺ cells from RFP⁺ Tregs was significantly higher in thymi than in spleens of the same recipients altogether suggesting that mature IL18R⁺ Tregs preferentially enter/remain in the thymus (*Figure 5c*, *Figure 5—figure supplement 2*).

To further validate whether IL18R itself is required for recirculation/retention of mature Tregs into the thymus, we compared the capacity of Tregs from *Il18r1* deficient mice (hereafter referred as *Il18r1*KO) and wild-type (wt) to populate the thymus. For this purpose, we isolated B220- and CD8-magnetically depleted splenocytes from *Il18r1*KO CD45.2 donors or IL18R sufficient CD45.1/CD45.2 heterozygous donors and co-transferred them into CD45.1 recipients (*Figure 5d*). As observed previously with RFP/GFP IL18R depleted Tregs, the ratio of wt to *Il18r1*KO Tregs was substantially higher in the thymus compared with that of the spleen (*Figure 5e*, *Figure 5—figure supplement 4*) and the frequency of IL18R⁺ from CD45.1/CD45.2 control cells was higher in thymi than in spleens of the same recipients (*Figure 5f*, *Figure 5—figure supplement 4*). These data, therefore, suggest that expression of IL18R on Tregs is essential for their effective recirculation into the thymus.

Finally, to provide additional evidence for the putative role of IL18R in Treg homing to the thymus, we analyzed the impact of IL18R deficiency (using *Il18r1*KO mice) on the frequency and numbers of different Treg subsets in the thymus. To this end, we utilized CD73 to identify mature Tregs (CD73⁺) from newly produced Tregs (CD73⁻) in the thymus as previously described in *Owen et al., 2019* (*Figure 1—figure supplement 1*). Specifically, we focused on total CD73⁺ Tregs as *Il18r1* deficiency precluded the identification of CD73⁺IL18R⁺ Tregs. Strikingly, flow cytometry analysis revealed that the counts of mature CD73⁺ Tregs in the thymus were significantly reduced in *Il18r1*KO mice (~50%; p=0.0079), while the numbers of thymic immature CD73⁻ Tregs or splenic Tregs

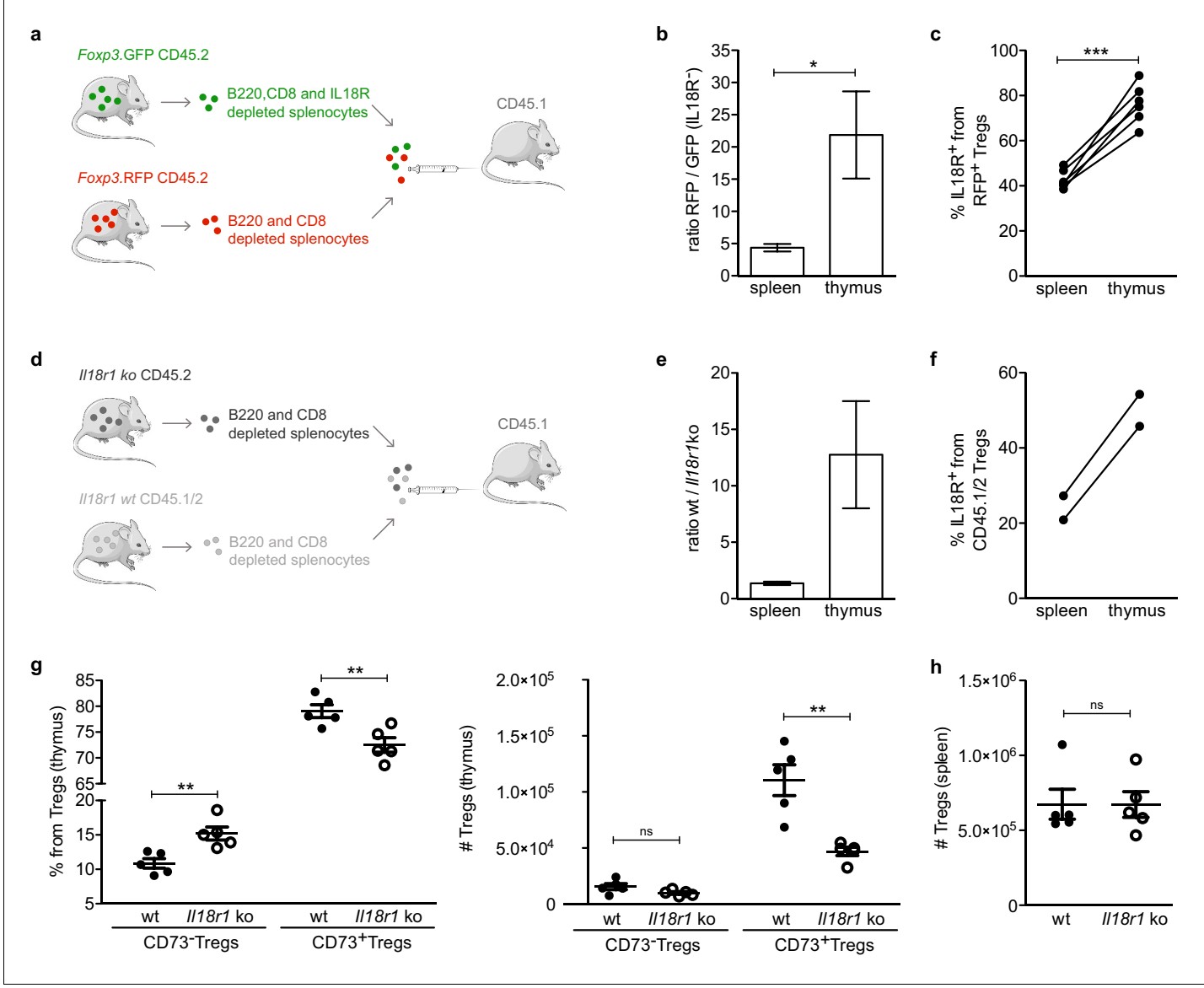

**Figure 5.** IL18R contributes to maintain mature Tregs in the thymus. (**a**) Schematic representation of the experiment in which IL18R depleted splenocytes from *Foxp3*. GFP mice were injected together with non-IL18R depleted splenocytes from *Foxp3*.RFP into CD45.1 recipients. (**b**) The ratio of RFP to GFP injected cells found in the spleen and thymus of recipient mice. (**c**) Frequency of IL18R⁺ Tregs from the *Foxp3*.RFP CD45.2 injected cells found in the spleen and thymus of recipient mice. (**d**) Schematic representation of the experiment in which *Il18r1*KO splenocytes were injected together with wild-type (wt) splenocytes into CD45.1 recipients. (**e**) The ratio of wt/*Il18r1*KO cells found in the spleen and thymus of recipient mice. (**f**) Frequency of IL18R⁺ Tregs from the wt injected cells found in the spleen and thymus of recipient mice. (**g**) Frequencies (left) and counts (right) of different Treg populations in the thymus of wt mice (solid dots) and *Il18r1*KO mice (empty dots). (**h**) Counts of Tregs in the spleen of wt and *Il18r1*KO mice. Each dot represents an individual mouse. The mean ± SEM (standard error of the mean) is shown. Data in (**b,c**) are from three independent experiments with a total of 6 biological replicates. (**e,f**) are from one experiment with a total of 2 biological replicates. Data in (**g,h**) are from one representative experiment with 12 total mice. Significant differences were determined in (**b,c,e,f**) using a paired t-test and in (**g,h**) using an unpaired t-test and indicated by asterisks *p<0.05, **p<0.01, ***p<0.001, ns: non-significant.

The online version of this article includes the following figure supplement(s) for figure 5:

**Figure supplement 1.** Purity of the injected splenocytes in the adoptive transfer experiments shown in *Figure 5*.

**Figure supplement 2.** Representative gating from *Figure 5a,b,c*.

**Figure supplement 3.** IL18R⁻ could give rise to IL18R⁺ Tregs.

**Figure supplement 4.** Representative gating from *Figure 5d,e,f*.

remained unchanged (*Figure 5g-h*). These data collectively demonstrate that expression of IL18R is essential for effective recirculation/retention of mature Tregs in the thymus.

## IL18 induces migration of Tregs into the thymus by upregulating CCR6

Previous reports showed that IL18 (the ligand for IL18R) can mediate chemotaxis of various immune cells including T, DC, and NK cells (*Gutzmer et al., 2003*; *Komai-Koma et al., 2003*; *Leung et al., 2001*) through an undefined mechanism. Therefore, to better understand whether IL18 signaling could directly mediate chemotaxis of mature Tregs, we tested the capacity of splenic Tregs to migrate toward IL18 in a transwell chemotaxis assay. Specifically, we introduced increasing doses of IL18, SDF-1α (the ligand of CXCR4, as a positive control), or medium (as a negative control) into the lower part of the transwell. Next, CD4-enriched splenocytes from *Foxp3*.GFP reporter mice (allowing identification of Tregs by GFP expression) were added on top of the transwell membrane and migration of Tregs across the filter toward the lower well was assessed after 30 min. Interestingly, the chemotactic index of IL18 was negligible for all tested physiologically active doses of IL18 (*Figure 6*), suggesting that IL18 does not directly chemoattract Tregs and that its contribution to Treg migration and retention in the thymus is likely indirect.

To further elucidate how IL18 signaling promotes recruitment/retention of mature Tregs in the thymus, we took an alternative and unbiased approach, based on the identification of transcriptional changes in Tregs in response to IL18 stimulation. To this end, we isolated splenocytes from *Foxp3*.GFP reporter mice and cultured them in vitro in the presence of CD3/CD28 activating beads together with increasing doses of IL18. After 48 and 96 hr, we sorted *Foxp3*.GFP$^+$ Tregs from the culture and analyzed them by bulk RNA sequencing. Differentially expressed genes common to both time-points are shown in *Figure 7a*. First, IL18 induced a transcriptional signature reminiscent of non-lymphoid tissue Tregs, characterized by elevated expression of *Asb2, Ttc39c, Il1rl1, Dgat2, Gpr15, Hopx* and lower expression of *Gpr83* and *Bcl2* (*Burzyn et al., 2013*; *Miragaia et al., 2019*; *Panduro et al., 2016*). Second, IL18 upregulated genes with anti-apoptotic function (*Traf1, Rnf157*) and downregulated genes with pro-apoptotic function (*Phlpp1, Trib3*) (*Arch et al., 1998*; *Gao et al., 2005*; *Matz et al., 2015*; *Ohoka et al., 2005*; *Wang et al., 1998*). Third, IL18 upregulated genes consistent with an effector phenotype that supports growth and proliferation, such as genes related to fatty acid biosynthesis (*Dgat2, Scd2, Hacd3*) (*Ikeda et al., 2008*; *Cle et al., 2008*) and *Hif1a*, which promotes glycolysis as well as migration of Tregs in glioblastoma (*Miska et al., 2019*). Lastly, IL18 stimulation also significantly upregulated *Ccr6*, encoding CCR6, the receptor of the chemokine CCL20 that has been previously shown to mediate Treg entry into the thymus (*Cowan et al., 2018*).

Finally, we analyzed the impact of IL18 stimulation on CCR6 surface expression on Tregs at a single cell resolution (*Figure 7b-c*). To this end, we used splenocytes from *Foxp3*.GFP reporter mice and cultured them in vitro without any stimuli or with CD3/CD28 activating beads in the absence or presence of IL18, and analyzed the CCR6 levels by flow cytometry. Indeed, the frequency of CCR6 expression in Tregs increased ~2-fold after 48 hr of IL18 stimulation (*Figure 7c*). The CCR6 increase was even bigger (~2.5-fold) after 96 hr, while CD3/CD28 activation alone did not impact CCR6 expression. The observed CCR6 increase in Tregs occurred in

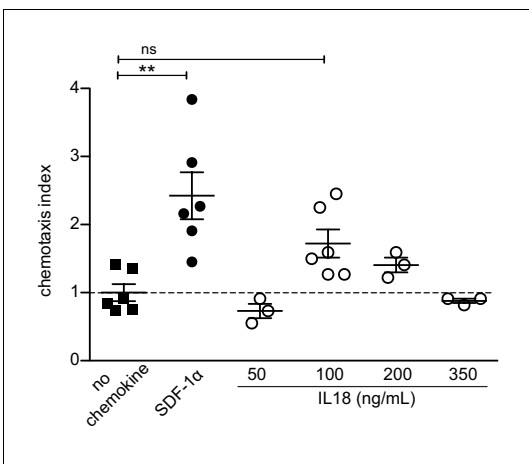

**Figure 6.** IL18 does not chemoattract Tregs. In vitro transwell chemotaxis assay showing migration of splenic Tregs in the presence of IL18 added to the lower wells at indicated doses. SDF-1α (CXCL12) was added at a concentration of 100 ng/mL and served as a positive control. The mean ± SEM (standard error of the mean) is shown. Data are from two independent experiments. The chemotaxis index was calculated as % migrated Tregs in the presence of stimulus/% migrated Tregs in the presence of medium alone (no chemokine). Significant differences were determined using ANOVA, corrected for multiple comparisons by the Bonferroni method and indicated by asterisks **p<0.01, ns: non-significant.

contrast to (CD25⁻FOXP3⁻) CD4 T cells, as IL18 had no significant impact on CCR6 expression in (CD25⁻FOXP3⁻) CD4 T cells. The increase in CCR6 expression in Tregs was indeed the cause of IL18R signaling as culturing of *Il18r1*KO splenocytes in the presence of IL18 did not increase CCR6 expression in *Il18r1*KO Tregs (*Figure 7—figure supplement 1*). Taken together, our data indicate that the IL18/IL18R pathway promotes survival of Tregs and induces migration of Tregs toward the thymus by upregulating CCR6.

## Discussion

The thymus is a primary lymphoid organ, critical for the establishment of immunocompetent T lymphocytes, which are then exported to the immune periphery for their safeguarding mission. However, over the past years, it has become clear that T cells, including regulatory T cells (Tregs), can not only leave the thymus, but can also recirculate back from the immune periphery (*Bosco et al., 2006*; *McCaughtry et al., 2007*; *Naparstek et al., 1982*; *Reinhardt et al., 2001*; *Thiault et al., 2015*; *Yang et al., 2014*; *Zhan et al., 2007*). Correspondingly, it has become well accepted that a pool of mature Tregs takes residence in the thymus (*Cowan et al., 2016*; *Thiault et al., 2015*; *Yang et al., 2014*); however, the identity of these cells, requirements for their establishment and function in the thymus is not completely understood.

This study provides important insights into the heterogeneity and biology of these mature Tregs in the thymus. Specifically, our study reveals that the mature Treg population in the thymus is composed of at least two distinct subsets, which can be distinguished by the surface expression of the receptor for the proinflammatory cytokine IL18. Based on the transcriptional signature, the IL18R⁺ mature Tregs are characterized with a more activated and tissue-resident phenotype in comparison to their IL18R⁻ counterparts or newly produced thymic Tregs. Namely, the IL18R⁺ Tregs highly resemble non-lymphoid tissue like effector Tregs, described recently in lymph nodes (*Miragaia et al., 2019*), which express various genes that are characteristic of tissue-resident Tregs, even though they are localized in a lymphoid tissue. In addition, IL18R⁺ Tregs are characterized by a transcriptional signature common to most tissue-resident T cell populations including low expression levels of *Klf2*, *S1pr1* (encoding S1P1), and *Ccr7* together with high expression levels of *Itgae* (encoding CD103) and *Prdm1* (encoding Blimp1). However, we did not observe lower expression of *Tcf7* (encoding TCF1) and higher expression of *Hobit*, typically required for establishing tissue residency of other lymphoid populations, such as CD8, ILC1, or NKT cells (reviewed in *Mackay and Kallies, 2017*). Together with their effector/tissue-resident-like molecular signature and their capacity to remain numerically stable in response to age- or stress-dependent thymus involution, our data strongly argue that the IL18R⁺ Tregs represent a stable thymus-resident population.

Moreover, our experimental data provide important novel insights into the molecular mechanisms underlying the homing of these cells into the thymus and established the critical role of IL18/IL18R signaling in this process. Specifically, our data demonstrate that IL18R⁺ Tregs have a significantly higher capacity to enter and/or remain inside the thymus than their IL18R⁻ or IL18RKO counterparts. This suggests that the thymic IL18R⁺ Tregs represent a recirculating population that may be maintained in the thymus by constant replenishment of Tregs from the periphery, in a similar manner to Tregs in the gut and the liver (*Luo et al., 2016*). Furthermore, the effector phenotype of the IL18R⁺ Tregs is also well in line with previous reports showing that activated T cells preferentially enter into the thymus (reviewed in *Hale and Fink, 2009*). In contrast, the origin of the IL18R⁻ Treg population is less clear. One possibility is that the IL18R⁻ Tregs may either originate from immature developing Tregs that are retained in the thymus before being released into the periphery or from IL18R⁺ Tregs that have recirculated to the thymus and have subsequently lost their IL18R expression. In either case, the fact that the IL18R⁻ Tregs express relatively high levels of *S1pr1* suggests that they are conditioned to exit the thymus. Moreover, our data demonstrate that once in the periphery, some of the IL18R⁻ Tregs can upregulate IL18R expression and recirculate to the thymus. Therefore, the recirculation of Tregs into the thymus and back to the periphery may be more dynamic than previously appreciated.

The involvement of IL18/IL18R signaling in facilitating homing of Tregs into the thymus is somewhat surprising, as IL18 is not a classic chemoattractant. Although several studies have demonstrated that it can promote the migration of various immune populations including Th1 cells, neutrophils, and dendritic cells (*Gutzmer et al., 2003*; *Komai-Koma et al., 2003*; *Leung et al.,*

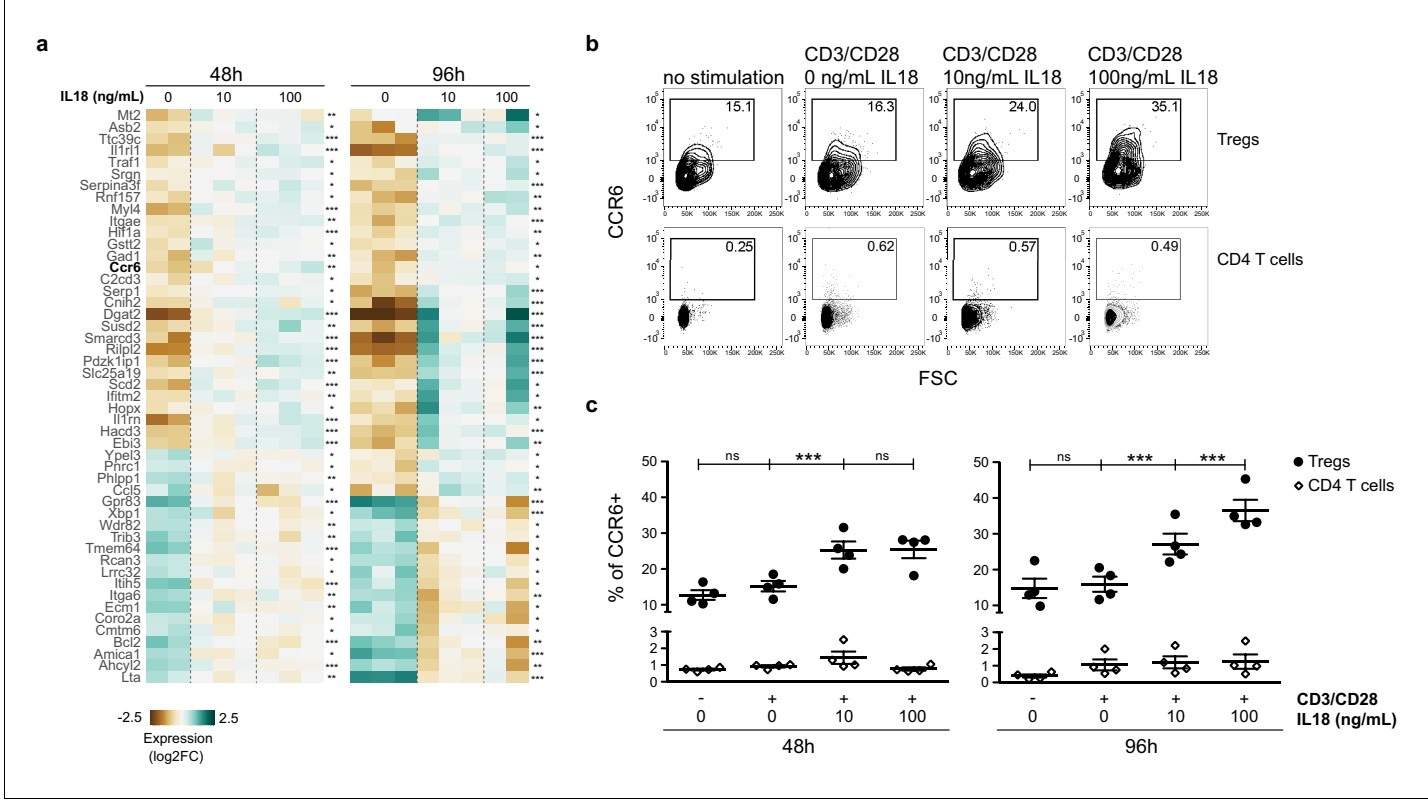

**Figure 7.** IL18/IL18R signaling upregulates CCR6. (**a**) A heatmap showing normalized expression of variable genes in splenic Tregs cultured with CD3/CD28 beads and absence/presence of IL18 (0, 10, 100 ng/mL). Selected genes have an adjusted p-value <0.1 in both 48 and 96 hr treatment versus control tests. Asterisks indicate adjusted p-values in each test *p<0.1, **p<0.01, ***p<0.001. (**b**) Representative flow cytometry plots showing CCR6 expression in CD25[+]FOXP3[+]Tregs (up) or CD25[-]FOXP3[-]CD4 T cells (down) cultured for 96 hr with no stimulation, CD3/CD28 beads, or CD3/CD28 beads plus absence/presence of IL18 (0, 10, 100 ng/mL). Numbers indicate the percentage of cells within each gate. (**c**) Frequency of CCR6 in CD25[+]FOXP3[+]Tregs (black dots) and CD25[-]FOXP3[-]CD4 T cells (empty rhombus) cultured for 48 or 96 hr with no stimulation, CD3/CD28 beads, or CD3/CD28 beads plus absence/presence of IL18 (0, 10, 100 ng/mL). Data are from two independent experiments with four biological replicates. Significant differences were determined using ANOVA, corrected for multiple comparisons using the Bonferroni method. For Tregs, differences are indicated by asterisks ***p<0.001, ns:non-significant. For CD4 T cells, ANOVA between conditions in both time-points was not significant.

The online version of this article includes the following figure supplement(s) for figure 7:

**Figure supplement 1.** CCR6 upregulation upon IL18 is dependent on IL18R signaling.

---

2001), the underlying mechanisms remained elusive. Importantly, our results demonstrate that IL18 does not directly chemoattract IL18R[+]Tregs in vitro, but rather it enhances Treg entry to the thymus indirectly, through induction of the key thymus-homing receptor *Ccr6*/CCR6. These results are in line with a previous study showing that in vitro stimulation with IL18 induces the expression of CCR6 in Th1 cells (*Facco et al., 2007*). In addition, as previously shown (*Kim et al., 2008*), IL18 upregulated the expression of the transcription factor *Hif1a*, which promoted migration of glycolysis-driven Tregs in a glioblastoma mouse model (*Miska et al., 2019*). According to our RNA sequencing data, IL18R signaling could also partially induce the non-lymphoid tissue like phenotype that IL18R[+] Tregs exhibit in the thymus, as it occurs in the gut (*Harrison et al., 2015*); and could also promote Treg survival, altogether contributing to the maintenance of IL18R[+] Tregs in the thymus.

Few advances have been made to understand the possible role of mature Tregs in the thymus. A previous report showed that recirculating Tregs in the thymus control the development of new Tregs by competing for IL2 availability (*Thiault et al., 2015*). However, this has been challenged by another study demonstrating that both *Aire*KO and *Ccr6*KO mice have reduced numbers of mature (*Rag.* GPF[−]) Tregs, with no concomitant increase of newly produced (*Rag.*GFP[+]) Tregs (*Cowan et al., 2018*). Our analysis of *Il18r1*KO mice thymi agrees with the conclusions of the latter study, as IL18R deficiency resulted in a substantial reduction of mature (CD73[+]) Tregs, but had no impact on the

newly produced (CD73⁻) Tregs. This discrepancy could potentially be explained by the possibility that the remaining fraction of mature Tregs is still able to limit Treg production. Nevertheless, other non-mutually exclusive functions of mature Tregs in the thymus could be proposed and may include (similarly to tissue-resident Tregs) control of tissue homeostasis.

Indeed, this idea is further supported by the fact that IL18R⁺ Treg frequency (as opposed to IL18R⁻ Treg and newly produced Treg) increases in parallel to either age- or stress-dependent involution and that IL18/IL18R pathway is critical for the maintenance of mature Tregs in the thymus. These observations, together with the capacity of IL18R to sense inflammation (*van der Veeken et al., 2016*; *Yasuda et al., 2019*), the fact that *Il18r1* is upregulated in specialized Tregs that promote tissue homeostasis in the lung (*Arpaia et al., 2015*), muscle (*Burzyn et al., 2013*), gut (*Harrison et al., 2015*), and visceral adipose tissue (*Vasanthakumar et al., 2015*), suggest altogether that IL18R⁺ Tregs could participate in thymus homeostasis as shown for Tregs in these specific tissues (*Panduro et al., 2016*). For instance, in the gut, IL18 signaling promotes Treg function preventing colitis (*Harrison et al., 2015*); while in the lung, IL18 signaling induces production of amphiregulin in Tregs, a growth factor promoting tissue repair and homeostasis (*Arpaia et al., 2015*). Interestingly, the IL18R⁺ thymic Tregs have relatively low expression of amphiregulin and *Il1rl1* (ST2), the receptor for IL33 alarmin, which is produced upon tissue damage. However, despite this difference, additional similarities exist between IL18R⁺ Tregs in the thymus and tissue-resident Tregs. For instance, the thymic IL18R⁺ Tregs (unlike their CD73⁺ IL18R⁻ or CD73⁻ counterparts) upregulate the expression of several genes that are associated with control of tissue homeostasis. These genes include *Tff1* reported to play a key role in the repair of gut epithelium (*Playford et al., 1996*), *Pdgfb* a potent mesenchymal cell mitogen that promotes tissue repair, and *Penk* (encoding proenkephalin A), which was shown to affect various biological processes including control of cell proliferation (*Cheng et al., 2010*; *Linner et al., 1995*; *Malendowicz et al., 2005*; *Zagon et al., 2011*), tissue repair (*Bigliardi-Qi et al., 2006*; *Yang et al., 2016*), or adipocyte metabolism/function (*Brestoff et al., 2015*).

In summary, our study provides a detailed molecular characterization of the major mature thymus Treg subsets and identifies a novel population of thymus recirculating Tregs, characterized by the expression of IL18R and resistance to age and stress-induced thymus involution. Moreover, our study uncovers the importance of IL18R in their trafficking and maintenance in the thymus, and identifies a key role of IL18 signaling in controlling a CCR6-CCL20-dependent migration axis used by IL18R⁺ Tregs to enter the thymus. These results not only provide an important basis for understanding the potential biological function(s) of this intriguing Treg population in the thymus, but also set the base to explore whether a similar IL18-dependent mechanism is critical for migration of Tregs (or other immune cell subsets) to other peripheral tissues. Given that IL18R is highly upregulated not only on peripheral Tregs, but also on activated T cells, NKT cells, or NK cells, these results might have important clinical implications in IL18-based immunotherapies currently being explored for pathologies such as chronic inflammation or cancer (*Mühl and Bachmann, 2019*).

## Materials and methods

### Mice

All C57/BL6 mice (C57/BL6JOlaHsd) were purchased from Envigo-Israel. *Rag*.GFP mice (FVB-Tg (Rag2-egfp)1Mnz/J; JAX stock 005688) were purchased from The Jackson laboratory and crossed to *Foxp3*.RFP mice (C57BL/6-Foxp3tm1Flv/J; JAX stock 008374) kindly provided by Y. Merbl to obtain *Foxp3*.RFP *Rag*.GFP mice. *Foxp3*.GFP mice (described in *Bettelli et al., 2006*) were kindly provided by G. Shakhar. CD45.1 mice (B6.SJL-Ptprca Pepcb/BoyJ; JAX strain 002014) were kindly provided by I. Schachar and crossed to C57/BL6 to obtain CD45.1/CD45.2 heterozygous mice. *Il18r1* KO mice (B6.129P2-Il18r1tm1Aki/J; JAX stock 004131) were purchased from The Jackson laboratory. All animals were housed under SPF conditions. All experiments were conducted in accordance with the guidelines for animal welfare approved by the Institutional Animal Care and Use Committee (IACUC) at Weizmann Institute of Science. Mice received standard rodent chow and autoclaved drinking water ad libitum.

## Thymocyte and splenocyte isolation

Thymi were collected in MACS buffer containing DPBS (BI 02-023-1A), 2% fetal bovine serum (FBS, BI 040071A), 5 mM EDTA and kept on ice. Single cell suspensions were prepared by smashing the thymi through a 40 µm strainer with a syringe plunger. Spleens were collected in RPMI medium 1640 (Gibco 21875–034), 5% fetal bovine serum (FBS, BI 040071A) and kept on ice. Single cell suspensions were prepared by smashing the spleens through a 40 µm strainer with a syringe plunger. Then red blood cells were lysed by incubating the single cell suspension in ammonium-chloride-potassium (ACK) lysis buffer (water, 150 mM $NH_4Cl$, 10 mM $KHCO_3$, 0.1 mM $Na_2EDTA$) for 6 min at room temperature, followed by two washes with RPMI 5% FBS.

## Cell staining and flow cytometry

After isolation, thymocytes and splenocytes were washed with MACS buffer (PBS, 2% FBS, 5 mM EDTA) and centrifuged. Pelleted cells (with the exception of CXCR4 and CCR6 staining) were then incubated for 30 min, 4°C, in a total volume of 100 µL with viability dye and fluorochrome-labeled monoclonal antibodies. Alternatively, for CXCR4 and CCR6 staining, pelleted cells were incubated 10 min at 37°C in a total volume of 50 µL with fluorochrome-labeled monoclonal antibodies for CXCR4 and CCR6. Then cells were incubated for 25 extra minutes at room temperature in an extra volume of 50 µL with the rest of fluorochrome-labeled monoclonal antibodies. After surface antibody staining, cells were washed with MACS buffer.

For intracellular detection, cells were fixed and permeabilized using the Truenuclear Buffer kit (Biolegend 424401) according to manufacturer's instructions. Cells were then incubated 30 min, at room temperature, in a total volume of 100 µL with fluorochrome-labeled monoclonal antibodies against FOXP3 and Ki67, then washed and finally resuspended in MACS buffer.

Stained cells were acquired in a flow cytometer within 2 hr after staining. Prior to acquisition cells were kept protected from light at 4°C. To determine cell counts, 50 uL of cell counting beads (Spherotech ACBP-100–10) were added to each sample before acquisition. Flow cytometry data were collected on a LSRII or FACS Aria III cytometers (BD Biosciences) and analyzed with Flow Jo software (Tree Star). At least 1 million of total cells were recorded. Dead cells were excluded on the basis of SSC-FSC profile and a viability marker.

## Intravascular staining

For intravascular staining, mice were intravenously injected with 8 µg of CD45 PerCPCy5.5 (clone 30-F11, Biolegend 103132) diluted in 200 µL of PBS. After 5 min, mice where euthanized, thymus and blood where collected. Blood was collected in a 20 mM EDTA coated tube and red blood cells were lysated with ACK lysis buffer prior to cell staining (as described above). Thymus was collected in MACS buffer (PBS, 2% FBS, 5 mM EDTA) in presence of 1.25 µg of CD45 unconjugated antibody (clone 30-F11, Biolegend 103101). A single cell suspension was prepared by smashing the organ prior to cell staining.

## Cell enrichment and sorting

For RNAseq experiments, 8–10 week old *Foxp3*.RFP *Rag*.GFP mice were used. Single cell suspensions from thymi and spleens were prepared by smashing the organs through a plunger using 40 µm strainer. For thymi, samples were CD8 depleted using magnetic beads (CD8a Ly-2 Microbeads Miltenyi 130-117-044). For spleens, samples were CD4 enriched using magnetic beads (CD4 L3t4 Microbeads 130-117-043) following the manufacturer's instructions. Alternatively, for RNAseq experiments from in vitro cultured cells (described in 'In vitro IL18 culture' section) cells were harvested from plates and washed in PBS twice.

The resulting single cells suspensions were then stained with fluorescently-labeled antibodies and DAPI (1:1000) 5 min, RT, before sorting by FACS Aria III cytometer. Cells were sorted directly unto lysis/binding buffer (Invitrogen 61012) and kept at −80°C until further processing of RNA.

## RNA isolation, sequencing and analysis

RNA was extracted using Dynabeads (Invitrogen 61012) following manufacturer's instructions. Sequencing libraries were prepared as described in *Jaitin et al., 2014* and sequenced by Illumina NextSeq platform, generating a total of 400M 50-base paired reads. We used the UTAP pipeline

(*Kohen et al., 2019*) to map these to the mouse genome (version mm10) and to calculate Unique Molecule Identifier (UMI) counts per gene. Significant differential expression was calculated using DESeq2 (*Love et al., 2014*), using either the Wald test (default DESeq2) for factors with two levels or a likelihood ratio test (LTR) for factors with more than two levels. The p-values were adjusted using Independent Hypothesis Weighting (IHW) as implemented in DESeq2. For visualization, gene expression (UMIs) was normalized dividing each value by the median expression of the gene, adding a regularization factor of 20 (corresponding approximately to the 50% quantile of the expression distribution).

### Irradiation and dexamethasone treatments

The 8–10-week-old mice were exposed to whole body sub-lethal X-ray irradiation (425cGy) or treated intraperitoneally with 20 mg/kg dexamethasone (Sigma 4902) (or ethanol vehicle). After 7–10 days, thymi and spleens were collected from treated mice.

### In vivo migration (adoptive transfer)

Donors were *Foxp3*.RFP and *Foxp3*.GFP mice or CD45.1/CD45.2 heterozygous and *Il18r1* deficient mice. Splenocytes from sex- and aged-matched donor mice were isolated as described earlier. In all samples, splenocytes were B220- CD8- depleted using magnetic beads. In some samples, splenocytes were also IL18R- depleted. In brief, splenocytes were incubated for 30 min, at 4°C with B220-APC (Biolegend 103212), CD8-APC (Biolegend 100712) in the presence or absence of IL18R-APC (eBioscience 175183–82). Then cells were washed with MACS buffer, and incubated with anti-APC Microbeads (Miltenyi 130-090-855) and magnetically depleted using LD columns (Miltenyi 130-042-901) according to manufacturer's instructions. The resulting populations were co-transferred intravenously into 10–15-week-old sex-matched CD45.1 recipient mice. After 12 days, thymi and spleens were collected from recipient mice.

### In vitro transwell chemotaxis assay

Chemotaxis toward IL18 was assessed as described previously (*Yadav et al., 2019*). In brief, splenocytes from *Foxp3*.GFP mice were depleted of CD8- and B220- cells (utilizing APC labeled antibodies and anti-APC magnetic beads described above in section 'in vivo migration (adoptive transfer)'). 2.5 million cells/mL were resuspended in binding medium (HBSS 1X containing 2 mg/mL BSA, 10 mM HEPES, pH 7.4 and supplemented with physiological concentrations of $CaCl_2$ and $MgCl_2$, that is 1 mM each) and added into the upper chamber of 6.5 mm transwells with 5 µm pore polycarbonate membrane inserts (Corning 3421). The bottom chambers were filled with binding medium alone or supplemented with recombinant mouse CXCL12 (SDF-1α, R and D Systems 460-SD) or IL18 (BioLegend 767002). Loading control with no chemokine was performed in parallel. Each condition was repeated in triplicates. After incubation for 30 min at 37°C, no $CO_2$, transwell inserts were removed, the cells from the bottom chambers were harvested and acquired using CytoFlex flow cytometer (Beckman Coulter). Tregs were identified by their GFP signal. Percentage of migration was calculated as: % migration = (migrated Tregs/loading control cells) x 100. The chemotaxis index was calculated as follows: chemotaxis index = % migration in the presence of stimulus/% migration in the presence of medium alone.

### In vitro IL18 culture

Splenocytes from wild-type and IL18RKO mice were isolated as described above. Splenocytes ($10^6$ cells/mL) were cultured in 96-flat bottom well plates in RPMI (Gibco 21875), 1% Pen/Strep (BI 030311B), 10% FBS (Gibco 012657), 1 mM sodium pyruvate (BI 030421B), 10 mM Hepes (BI 030251B), and 10 µM β-mercaptoethanol (Gibco 31350–010). In addition, cells were cultured with or without CD3/CD28 beads (Miltenyi 130-095-925; 4:1 ratio cell to bead), and 10 ng/mL or 100 ng/mL recombinant mouse IL18 (Biolegend 767002). Cells were cultured at 37°C, 5% $CO_2$ for 48 or 96 hr. Afterward, cells were harvested and stained as described earlier to proceed with Treg sorting for RNA sequencing or CCR6 evaluation by flow cytometry.

## Statistical analysis

Except for the RNA-seq data (mentioned earlier), all statistical analyses were performed using GraphPad Prism 5.0. Comparisons between two groups were performed using a t-test, between multiple groups using ANOVA and between two organs within the same recipient using a paired t-test. Correlation coefficients (r) were calculated using the Spearman's rank correlation test. Corrected *p*-values (p) <0.05 were considered significant.

## Acknowledgements

We thank all members of the Abramson laboratory for many helpful discussions. We thank Dr. Y. Merbl for *Foxp3*.RFP mice, Dr. I. Shachar for CD45.1 mice, and Dr. G. Shakhar for *Foxp3*.GFP mice. We thank the Weizmann flow cytometry unit for their support with cell sorting experiments. We thank the animal facility personnel and the Abramson lab genotyping team for help with maintaining mice strains.

Research in the Abramson laboratory is kindly supported by the European Research Council (ERC-2016-CoG-724821), Israel Science Foundation (1796/16), Sy Syms Foundation, Bill and Marika Glied and Family Fund, Wohl Biology Endowment Fund, Erica Drake Fund, The Enoch Foundation, Ruth and Samuel David Gameroff Family Foundation, and Lilly Fulop Fund for Multiple Sclerosis Research. CP was supported by Weizmann-La Caixa fellowship and Weizmann-Dean of faculty fellowship.

## Additional information

### Funding

| Funder | Grant reference number | Author |
|---|---|---|
| "la Caixa" Foundation | Postdoctoral Fellowship | Cristina Peligero-Cruz |
| European Research Council | ERC-2016-CoG-724821 | Jakub Abramson |
| Israel Science Foundation | 1796/16 | Jakub Abramson |
| Sy Syms Foundation | | Jakub Abramson |
| Bill and Marika Glied and Family Fund | | Jakub Abramson |
| American Committee for the Weizmann Institute of Science | | Jakub Abramson |
| Erica Drake Fund | | Jakub Abramson |
| The Enoch Foundation | | Jakub Abramson |
| Ruth and Samuel David Gameroff Family Foundation | | Jakub Abramson |
| Lilly Fulop Fund for Multiple Sclerosis Research | | Jakub Abramson |

The funders had no role in study design, data collection and interpretation, or the decision to submit the work for publication.

### Author contributions

Cristina Peligero-Cruz, Conceptualization, Software, Formal analysis, Supervision, Investigation, Writing - original draft, Project administration, Writing - review and editing; Tal Givony, Noam Kadouri, Shir Nevo, Francesco Roncato, Investigation; Arnau Sebé-Pedrós, Software, Formal analysis, Writing - review and editing; Jan Dobeš, Yael Goldfarb, Investigation, Writing - review and editing; Ronen Alon, Resources, Writing - review and editing; Jakub Abramson, Conceptualization, Supervision, Funding acquisition, Writing - original draft, Project administration, Writing - review and editing

## Author ORCIDs

Cristina Peligero-Cruz (iD) https://orcid.org/0000-0001-7465-5008
Jan Dobeš (iD) http://orcid.org/0000-0003-1853-1603
Jakub Abramson (iD) https://orcid.org/0000-0002-1745-8996

## Ethics

Animal experimentation: All animal procedures were conducted in strict accordance with the guidelines for animal welfare approved by the Institutional Animal Care and Use Committee (IACUC) at Weizmann Institute of Science (project numbers 36220617-2, 11370219-1,11380219-3).

## Decision letter and Author response

Decision letter https://doi.org/10.7554/eLife.58213.sa1
Author response https://doi.org/10.7554/eLife.58213.sa2

# Additional files

## Supplementary files

- Transparent reporting form

## Data availability

Sequencing data have been deposited in NCBI's Gene Expression Omnibus and are accessible through GEO Series accession number GSE153155.

The following dataset was generated:

| Author(s) | Year | Dataset title | Dataset URL | Database and Identifier |
|---|---|---|---|---|
| Peligero-Cruz C, Abramson J | 2020 | IL18 signaling promotes homing of mature Tregs into the thymus | https://www.ncbi.nlm.nih.gov/geo/query/acc.cgi?acc=GSE153155 | NCBI Gene Expression Omnibus, GSE153155 |

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

# Appendix 1

**Appendix 1—key resources table**

| Reagent type (species) or resource | Designation | Source or reference | Identifiers | Additional information |
|---|---|---|---|---|
| Antibody | anti-CD3 APC (rat monoclonal) | Biolegend | Cat#100236 | FACS (1uL/test) |
| Antibody | anti-CD3 PerCP/Cy5.5 (rat monoclonal) | Biolegend | Cat#100218 | FACS (1,5uL/test) |
| Antibody | anti-CD4 PB (rat monoclonal) | Biolegend | Cat#100428 | FACS (1uL/test) |
| Antibody | anti-CD4 PECy7 (rat monoclonal) | Biolegend | Cat#100422 | FACS (0,5uL/test) |
| Antibody | anti-CD8a PB (rat monoclonal) | Biolegend | Cat#100725 | FACS (0,5uL/test) |
| Antibody | anti-CD8a APC (rat monoclonal) | Biolegend | Cat#100712 | FACS (0,5uL/test) |
| Antibody | anti-CD25 PE (rat monoclonal) | Biolegend | Cat#101904 | FACS (2uL/test) |
| Antibody | anti-CD25 BV711 (rat monoclonal) | Biolegend | Cat#102049 | FACS (0,5uL/test) |
| Antibody | anti-CD45 PerCP/Cy5.5 (rat monoclonal) | Biolegend | Cat#103132 | $8\mu g$ in $200\mu L$ PBS iv/mouse |
| Antibody | anti-CD45 unconjugated (rat monoclonal) | Biolegend | Cat#103101 | $1.25\mu g$ in 2mL Macs/organ |
| Antibody | anti-CD45.1 APC/Cy7 (mouse monoclonal) | Biolegend | Cat#110715 | FACS (2uL/test) |
| Antibody | anti-CD45.2 PerCP/Cy5.5 (mouse monoclonal) | Biolegend | Cat#109827 | FACS (1uL/test) |
| Antibody | anti-CD45.2 APC (mouse monoclonal) | Biolegend | Cat#109814 | FACS (0,5uL/test) |
| Antibody | anti-CD45R(B220) APC (rat monoclonal) | Biolegend | Cat#103212 | FACS (1uL/test) or $6\mu L$ (in total$600\mu L$)/spleen for depletion |
| Antibody | anti-CD62L APC/Cy7 (rat monoclonal) | Biolegend | Cat#104427 | FACS (1uL/test) |
| Antibody | anti-CD69 APC/Cy7 (Armenian Hamster monoclonal) | Biolegend | Cat#104525 | FACS (1uL/test) |
| Antibody | anti-CD73 APC (rat monoclonal) | Biolegend | Cat#127210 | FACS (0,3uL/test) |
| Antibody | anti-CD73 PE (rat monoclonal) | Biolegend | Cat#127206 | FACS (0,3uL/test) |

*Appendix 1—key resources table continued*

| Reagent type (species) or resource | Designation | Source or reference | Identifiers | Additional information |
|---|---|---|---|---|
| Antibody | anti-CD103 APC (Armenian Hamster monoclonal) | Biolegend | Cat#121413 | FACS (1uL/test) |
| Antibody | anti-CD184 (CXCR4) APC (rat monoclonal) | Biolegend | Cat#146507 | FACS (0,5uL/test) |
| Antibody | anti-CD196 (CCR6) BV605 (Armenian Hamster monoclonal) | Biolegend | Cat#129819 | FACS (2uL/test) |
| Antibody | anti-CD218a (IL18Ra) PerCPF710 (rat monoclonal) | eBiosciences | Cat#465183-80 | FACS (1uL/test) |
| Antibody | anti-CD218a (IL18Ra) APC (rat monoclonal) | eBiosciences | Cat#175183-82 | FACS (1uL/test) |
| Antibody | anti-CD279 (PD-1) PECy7 (rat monoclonal) | Biolegend | Cat#135215 | FACS (2uL/test) |
| Antibody | anti-Foxp3 AF488 (rat monoclonal) | Biolegend | Cat#126406 | FACS (1uL/test) |
| Antibody | anti-Ki67 BV605 (rat monoclonal) | Biolegend | Cat#652413 | FACS (1uL/test) |
| Antibody | anti-Ly6-G/Ly6-C APC (rat monoclonal) | Biolegend | Cat#108412 | FACS (2uL/test) |
| Other | Fixable viability dye e506 | eBioscience | Cat#65-0866-18 | FACS (0,1uL/test) |
| Other | Dapi | Merck | Cat#10236276001 | 1:1000 RT, 5min, no wash for sorting alive cells |
| Chemical compound, drug | Dexamethasone | Sigma | Cat#4902 | |
| Peptide, recombinant protein | CXCL12 (SDF-1$\alpha$) | R&D Systems | Cat#460-SD | |
| Peptide, recombinant protein | IL18 | Biolegend | Cat#767002 | |
| Commercial assay or kit | Truenuclear Buffer kit | Biolegend | Cat#424401 | |
| Commercial assay or kit | CD8a Ly-2 Microbeads | Miltenyi | Cat#130-117-044 | |
| Commercial assay or kit | CD4 L3t4 Microbeads | Miltenyi | Cat#130-117-043 | |
| Commercial assay or kit | anti-APC Microbeads | Miltenyi | Cat#130-090-855 | |

*Appendix 1—key resources table continued*

| Reagent type (species) or resource | Designation | Source or reference | Identifiers | Additional information |
|---|---|---|---|---|
| Commercial assay or kit | LD columns | Miltenyi | Cat#130-042-901 | |
| Commercial assay or kit | Lysis/Binding buffer | Invitrogen | Cat#61012 | |
| Commercial assay or kit | Dynabeads | Invitrogen | Cat#61012 | |
| Commercial assay or kit | CD3/CD28 beads | Miltenyi | Cat#130-095-925 | |
| Strain, strain background (*M. musculus*) | C57/BL6 mice (C57/BL6JOlaHsd) | Envigo-Israel | | |
| Strain, strain background (*M. musculus*) | *Rag*.GFP mice (FVB-Tg (Rag2-egfp)1Mnz/J) | The Jackson laboratory | JAX stock 005688 | |
| Strain, strain background (*M. musculus*) | *Foxp3*.RFP mice (C57BL/6-Foxp3tm1Flv/J) | The Jackson laboratory Y.Merbl | JAX stock 008374 | |
| Strain, strain background (*M. musculus*) | *Foxp3*.GFP mice | G. Shakhar **Bettelli et al., 2006** | | |
| Strain, strain background (*M. musculus*) | CD45.1 mice (B6.SJL-Ptprca Pepcb/BoyJ) | The Jackson laboratory I. Schachar | JAX stock 002014 | |
| Strain, strain background (*M. musculus*) | *Il18r1* KO mice (B6.129P2-Il18r1tm1Aki/J) | The Jackson laboratory | JAX stock 004131 | |
| Software, algorithm | Flow Jo | Treestar | www.flowjo.com | |
| Software, algorithm | Graphpad Prism | Graphpad | www.graphpad.com | |
| Other | counting beads | Spherotech | Cat#ACBP-100-10 | |
| Other | 6.5mm transwells, 5μm pore inserts | Corning | Cat#3421 | |

