## [Decision Letter]

**Acceptance summary:**

This work represents an important advance because it identifies a novel population of mature thymic T regs, and demonstrates a role for the IL-18/IL-18R axis in Treg recirculation to the thymus. In addition to shedding new light on an important and poorly understood topic, the similarities between the recirculating IL18R^+^ Tregs and tissue-resident T cell are intriguing.

**Decision letter after peer review:**

Thank you for submitting your article "IL18 signaling promotes homing of mature Tregs into the thymus" for consideration by *eLife*. Your article has been reviewed by three peer reviewers, one of whom is a member of our Board of Reviewing Editors, and the evaluation has been overseen by Tadatsugu Taniguchi as the Senior Editor. The following individuals involved in review of your submission have agreed to reveal their identity: Graham Anderson (Reviewer #3).

The reviewers have discussed the reviews with one another and the Reviewing Editor has drafted this decision to help you prepare a revised submission.

Summary:

Previous studies have shown that a portion of thymic Tregs are relatively mature and arise from recirculation or long-term residence in the thymus. The current study examines heterogeneity within the mature thymic Treg population, revealing that IL15R marks a subpopulation of mature Tregs whose numbers are remarkably resistant to age or irradiation induced thymic involution. The authors characterize the cells based on detailed gene expression, flow cytometry, Foxp3 and Rag2 reporter labeling, and cell transfer experiments. They show that IL-18R is functionally important to enable the cells to home to the thymus upon injection into a new host. They also show that one role for IL18R is to induce CCR6 expression, which has previously been shown to be important in Treg recirculation back to the thymus. Overall, the experiments are well performed and convincing. Given that Treg recirculation to the thymus remains enigmatic, and that little is known about the IL-18/IL-18R axis in the thymus in general, this study represents a significant advance.

Revisions:

1) The authors discuss 2 possible functions for IL18R^+^ mature thymic Tregs: regulating new Treg development and tissue homeostasis. Are there any indications that the IL18R^+^ differ in their ability to perform these functions compared to mature IL18R^-^ or newly developed Tregs? The manuscript could be improved by an expanded consideration of this topic.

2) While very well written with graceful handling of the background literature, the manuscript does not indicate clearly what "crossover" scientific lessons emerge from the results for a broad audience in *eLife*, as opposed to an expert audience of immunologists.

3) Throughout the study, the authors use terms such as thymus-resident, tissue-resident, recirculating to describe the Rag2GFP-IL18R^+^ Treg present in thymus. Most of the evidence the authors provide, including in vivo cell transfer studies, indicate that these cells are recirculating. Tissue/thymus resident suggests that these cells have not left the thymus, and have been retained in the thymus for long periods. To clarify this issue, the authors should consider referring to these cells throughout as “thymus recirculating Treg”, which in the absence of data to demonstrate their long-term intrathymic persistence, seems to be more accurate.

---

## [Author Response]

Revisions:1) The authors discuss 2 possible functions for IL18R^+^ mature thymic Tregs: regulating new Treg development and tissue homeostasis. Are there any indications that the IL18R^+^ differ in their ability to perform these functions compared to mature IL18R^-^ or newly developed Tregs? The manuscript could be improved by an expanded consideration of this topic.

We appreciate this suggestion and in the revised manuscript we try to present this topic in a clearer and more detailed way. Specifically, in the Discussion section (paragraph six), we provide more examples of genes that are highly expressed in the IL18R^+^ subset (vis-à-vis IL18R^-^ or *Rag.*GFP^+^) and are associated with tissue homeostasis.

2) While very well written with graceful handling of the background literature, the manuscript does not indicate clearly what "crossover" scientific lessons emerge from the results for a broad audience in eLife, as opposed to an expert audience of immunologists.

We would like to thank the reviewer for encouraging us to propose a broader significance and impact that stem from our findings. In the revised manuscript we extended the Discussion in the concluding paragraph to better contextualize the relevance of our results for a broader audience (paragraph seven).

3) Throughout the study, the authors use terms such as thymus-resident, tissue-resident, recirculating to describe the Rag2GFP-IL18R^+^ Treg present in thymus. Most of the evidence the authors provide, including in vivo cell transfer studies, indicate that these cells are recirculating. Tissue/thymus resident suggests that these cells have not left the thymus, and have been retained in the thymus for long periods. To clarify this issue, the authors should consider referring to these cells throughout as “thymus recirculating Treg”, which in the absence of data to demonstrate their long-term intrathymic persistence, seems to be more accurate.

We fully agree that “thymus recirculating Tregs” is probably a clearer and more accurate term to refer to this population. Therefore, in the revised manuscript we modified the terminology and refer to these cells as “thymus recirculating Tregs” throughout the study. Please see changes in the Abstract, Introduction and final paragraph of the Discussion.